# A Fish of Multiple Faces, Which Show Us Enigmatic and Incredible Phenomena in Nature: Biology and Cytogenetics of the Genus *Carassius*

**DOI:** 10.3390/ijms23158095

**Published:** 2022-07-22

**Authors:** Martin Knytl, Adrian Forsythe, Lukáš Kalous

**Affiliations:** 1Department of Cell Biology, Faculty of Science, Charles University, 12843 Prague, Czech Republic; 2Department of Ecology and Genetics, Evolutionary Biology Center, Uppsala University, 75236 Uppsala, Sweden; adrian.forsythe@ebc.uu.se; 3Department of Zoology and Fisheries, Faculty of Agrobiology, Food and Natural Resources, Czech University of Life Sciences Prague, 16521 Prague, Czech Republic; kalous@af.czu.cz

**Keywords:** hybridization, sexuality, asexuality, biotype, species, sex determination, ploidy level, *Carassius auratus* complex

## Abstract

Sexual vs. asexual reproduction—unisexual vs. bisexual populations—diploid vs. polyploid biotypes—genetic vs. environmental sex determination: all these natural phenomena are associated with the genus of teleost fish, *Carassius*. This review places emphasis on two *Carassius* entities with completely different biological characteristics: one globally widespread and invasive *Carassius gibelio*, and the other *C. carassius* with a decreasing trend of natural occurrence. Comprehensive biological and cytogenetic knowledge of both entities, including the physical interactions between them, can help to balance the advantages of highly invasive and disadvantages of threatened species. For example, the benefits of a wide-ranged colonization can lead to the extinction of native species or be compensated by parasitic enemies and lead to equilibrium. This review emphasizes the comprehensive biology and cytogenetic knowledge and the importance of the *Carassius* genus as one of the most useful experimental vertebrate models for evolutionary biology and genetics. Secondly, the review points out that effective molecular cytogenetics should be used for the identification of various species, ploidy levels, and hybrids. The proposed investigation of these hallmark characteristics in *Carassius* may be applied in conservation efforts to sustain threatened populations in their native ranges. Furthermore, the review focuses on the consequences of the co-occurrence of native and non-native species and outlines future perspectives of *Carassius* research.

## 1. Introduction

The genus *Carassius* (Nilsson, 1832) belongs to a monophyletic group of paleotetraploids from the Cyprinini tribe (sensu [1]), within the family Cyprinidae (Actinopterygii, Teleostei). Two valid, morphologically distinct and phylogenetically diverged species of *Carassius* have been clearly described, including crucian carp (*Carassius carassius* Linnaeus, 1758) and white crucian carp, also known as “gengoro-buna” (*C. cuvieri* Temminck and Schlegel, 1846). However, despite this distinction between several species of *Carassius* in the literature, taxonomic classification is often inconsistent and often incorrectly referred to by authors (e.g., [2,3]) as various different taxa, species, or holotypes.

## 2. Taxonomy of the Genus *Carassius*

Distinct morphological characteristics have emerged in *C. carassius* and *C. cuvieri*. *Carassius carassius* can be distinguished from all other *Carassius* by the convex angle of the upper edge of the dorsal fin, a black dot at the base of the caudal peduncle, and whitish peritoneum [4,5]. Despite the relatively clear differences in morphology, the use of the name *C. carassius* has been disputed. Without the use of modern molecular techniques, some authors have previously classified *C. carassius* as Japanese Funa or “Black-Funa” (sensu [6]), and an ancestor of goldfish [6,7,8]. However, these classifications are incorrect, based on previous phylogeographical studies that show *C. carassius* (sensu Linnaeus, 1758) as being native to many European freshwaters, with no records of occurrence in Japan [3,9,10,11]. *Carassius cuvieri* is phylogenetically distinct from *C. carassius* and all other *Carassius* species, representing a basal diverged clade [12,13] with morphological differences that do not correspond with levels of genetic differentiation [14]. Distinct characteristics of *C. cuvieri* include an increased number of gill rakers and endemic to *C. cuvieri* in Lake Biwa, Japan [2].

Other *Carassius* species and subspecies belong to the *C. auratus* complex, which is a biologically diverse group [15] with a wide geographical distribution [10,12], different levels of ploidy [16], various modes of reproduction [17], and distinct modes of sex determination [18]. Most notably, silver carp (gibel carp, *C. auratus gibelio*, Bloch, 1782) and *C. auratus auratus* (Linnaeus, 1758), are globally widespread and invasive to parts of mainland Eurasia, where they are highly abundant [5]. Additional subspecies occur mainly in the Japanese archipelago and Taiwan (some of them as endemic subspecies): ginbuna (*C. auratus langsdorfii* Temminck and Schlegel, 1846), nigorobuna (*C. auratus grandoculis* Temminck and Schlegel, 1846), okinbuna (*C. auratus buergeri* Temminck and Schlegel, 1846), nagabuna (*C. auratus* subsp. 1) and kinbuna (*C. auratus* subsp. 2) [2]. Some ichthyologists have considered *C. auratus*, *C. gibelio*, and *C. langsdorfii* as separate independent species [3,5].

The most well-known member of *Carassius*, the goldfish (*C. auratus*), commonly persists in an array of colorful varieties and body morphologies. However, much contention surrounds the taxonomic classification of this species. Identification of *C. auratus* has been inconsistent for a number of reasons, with some proposing that difficulties in classification are due to: (i) the possibility of multiple independent origins [12], (ii) gene flow from domesticated and feral populations [19], (iii) anthropogenic translocation [12], or (iv) inter-taxa hybridization [20]. Typically, the name *C. auratus* involves either a domesticated form of the goldfish or the entire *C. auratus* complex including *C*. (*auratus*) *gibelio*, *C*. (*auratus*) *auratus*, *C*. (*auratus*) *langsdorfii*, *C*. (*auratus*) *grandoculis*, *C*. (*auratus*) *buergeri*, *C*. (*auratus*) subsp. 1, and *C*. (*auratus*) subsp. 2. It is difficult to assign distinct “species” or “subspecies” based on morphological characteristics alone [14], as many exceptions between genetically uniform taxa exist in the *Carassius auratus* complex (e.g., frequent natural hybridizations) [21]. Even molecular genetic analysis and reconstruction of phylogenetic trees cannot reliably distinguish species from subspecies [10,12]. Inconsistent identification of *Carassius* species has persisted since the first identification of *Cyprinus carassius* and *Cyprinus gibelio* by Bloch, 1782. In general, *Cyprinus* and *Carassius* are recognized as two separate genera. In part of the natural history Museum für Naturkunde der Humboldt Universität zu (Berlin, Germany), these two specimens were clearly morphologically indistinguishable from *C. carassius*. In this case, the *Cyprinus gibelio* type specimen was likely lost and replaced by a specimen of *C. carassius* [3,22], making it difficult to judge whether the original identification of *Cyprinus gibelio* by Bloch represents a wholly different species or a hybrid between *Cyprinus* and *Carassius*. Because a holotype is missing in Bloch’s specimen collection, Kalous et al. [3] described a neotype for *C. gibelio* and concluded based on mitochondrial cytochrome *b* sequences that an additional taxon has likely been grouped under the species *C. gibelio*. This finding indicates the polyphyletic origin of *C. gibelio* and sparks doubt if the sister taxon still belongs to *C. gibelio*. Other alternative classifications have been proposed, such as *C. gibelio gibelio* or *C. carassius gibelio* [23], but neither of these classifications is widely used.

In this review, we will be keeping with the concept that the *C. auratus* complex includes all *Carassius* taxa, except for *C. carassius* and *C. cuvieri*, which will be treated as independent species. Other members of the *C. auratus* complex will be used under the scientific names *C. auratus*, *C. gibelio*, and *C. langsdorfii*. *Carassius auratus* will be applied only in the context of domesticated/feral goldfish.

## 3. Early Cytogenetic Studies of *Carassius* Karyotypes

A landmark cytogenetic study was published by Makino [24] in 1934, which first defined the chromosome number and karyotype of the genus *Carassius*. This work led to the first description of diploidy in goldfish (*C. auratus*) 2n=4x=94 from Japan, where “*n*” refers to a number of chromosomes in each gamete of extant species, and “*x*” refers to the number of chromosomes in a gamete of the most recent diploid ancestor of the extant species. Descriptions of the karyotypic diversity in *Carassius* were further extended by Makino [6], who reported the karyotypes of 13 colorful forms of goldfish as well as *C. carassius*. From this work, it can be concluded that misidentified *C. carassius* belongs to the taxon *C. auratus* due to the Japanese origin of the original samples. The karyotype of *C. gibelio* was first recorded by Cherfas [25], revealing that this species is represented by diploid and triploid individuals with chromosome numbers 2n=4x=94 and 3n=6x=141, respectively. A review of publications exploring the diverse karyotpes of *Carassius* is summarised in Table 1, with descriptions of ploidy in *C. auratus*, *C. gibelio*, *C. langsdorfii*, *C. cuvieri*, *C. buergeri*, and *C. grandoculis*. Additional information on the numbers of chromosomes and karyotype formulas for diploid *C. auratus*, *C. carassius*, and *C. gibelio* have previously been reviewed in Knytl and Fornaini ([26], Table 3), and therefore, they are not included in this review. The standardized karyotypes for diploids *C. auratus*, *C. carassius*, and *C. gibelio* have also been recently established [26].

*Carassius carassius*, *C. cuvieri*, and *C. grandoculis* are exclusively diploid with 100 chromosomes (2n=4x=100) [48,49,50]. *Carassius auratus*, *C. gibelio*, and *C. langsdorfii* form diploid (2n=4x=100), triploid (3n=6x≈150) and tetraploid biotypes (4n=8x≈200) [41,45,51,52]. Tetraploid *Carassius* have been usually generated by interspecific crossing of a triploid *Carassius* female with diploid heterospecific (i.e., from another related species) male [53,54], or naturally occurring pure *Carassius* tetraploids have not been cytogenetically described. Biologically diploid *Carassius* biotypes are interestingly considered evolutionary paleo-tetraploids [49,55] because most members of the family Cyprinidae contain 25 chromosomes in each gamete (50 chromosomes in a somatic cell).

## 4. Synergy of Conventional Chromosome Banding and Ribosomal Fluorescent In Situ Hybridization

Giemsa staining (G-banding) is the most widely used cytogenetic technique and is commonly used to visualize DNA segments or entire chromosomes by binding to AT-rich regions with high affinity (intense AT bands) and low affinity to GC-rich regions (negative GC bands). Comparatively, C- and CMA${}_{3}$-banding are staining techniques useful for labeling GC-rich chromosomal loci and constitutive heterochromatin. The above-mentioned banding techniques have served as useful cytogenetic tools, allowing for efficient and precise analyses of karyotypes in various animals. In particular, techniques beyond the routine Giemsa staining have promoted the development of cytogenetics of *Carassius* species to a great extent, especially at the end of the last century [30,35,48,56]. Specific chromosomal banding patterns are helpful in the identification of homologous chromosome pairs in a karyotype, just as morphological characteristics of chromosomes (length, arm ratio) are useful in distinguishing individual chromosomes between each other, which are not always evident [57]. Conventional banding approaches will, therefore, continue to be an integral and additional part of future cytogenetic research.

Conventional banding techniques together with more advanced molecular cytogenetics (e.g., in situ hybridization) are widely utilized in many species and may determine a tempo of karyotype evolution focused on repetitive ribosomal DNA (rDNA) loci, microsatellite motifs, ratio of parental genomes within a hybrid genome etc. [58,59,60]. The fluorescent in situ hybridization (FISH) method has a diverse utilization in a study of evolutionary processes due to many modifications focused on various types of targeted nucleotide sequences used as probes in various animal species [61,62,63]. The FISH method, in which rDNA probes are hybridized on chromosomal DNA, reveals exact chromosomal positions where the rDNAs are situated. Regions containing rDNA can be easily amplified using PCR and labeled with specific nucleotides using labeling PCR [64] or nick translation [65]. These regions encoding rRNA are arranged into tandemly repeated units: major *45S* locus composed of *18S*, *5.8S*, and *28S* genes, and minor *5S* locus, is formed independently. The *45S* locus creates nucleolar secondary constriction, which is called nucleolar organizer region (NOR) present during the M phase of the cell cycle, thus detectable using an rDNA FISH technique (as well as using Ag-NOR and CMA${}_{3}$ staining). The ribosomal genes are highly conserved in terms of sequence similarity across species [66]. Therefore, one identical probe can be used for rDNA FISH for multiple species analysis, e.g., Spoz et al. [50] used the human and loach (genus *Cobitis* Linnaeus, 1758) *5S* and *28S* rDNA probes for hybridization with *C. carassius* chromosomal DNA; Knytl et al. [67] used pike (*Esox lucius* Linnaeus, 1758) *5S* rDNA probes for hybridization on *C. carassius* chromosomes. The number of positive *5S* rDNA loci within the *Carassius* karyotype is not consistent because of the high number of these loci. Spoz et al. [50], Knytl et al. [67] and Knytl and Fornaini [26] found from 8 to 18 positive *5S* rDNA loci in *C. carassius*. Knytl and Fornaini [26] revealed ten *5S* rDNA signals in diploid *C. auratus* and *C. gibelio*. Zhu et al. [68] revealed 9-21 positive *5S* rDNA loci in triploid *C. gibelio*, Knytl et al. [67] detected 27 positive *5S* rDNA signals in triploid *Carassius*. The number of intensively stained *5S* rDNA loci agreed on the ploidy level in *Carassius*, i.e., two intensively fluorescent loci in diploids, three in triploids [26,67,68] and four in tetraploids [69]. The number of NORs is usually four within diploid *Carassius* biotypes [49,50,67,70], but the polyploid *Carassius* biotypes showed different numbers of NORs. Triploid *C. gibelio* has four [67] or five NORs [71,72], tetraploid *Carassius* bears six [71,72] or eight NORs [70]. The number, size and localization of rDNA patterns in *Carassius* are not consistently specific for each species, but the intensity of these rDNA positive signals could serve as a specific marker for ploidy level determination and for determination of how many times the genome has been duplicated (Figure 1).

## 5. Asexuality, Unisexuality, and Sexuality

One of the other characteristics of the genus *Carassius* is the ability to reproduce asexually [25]. Gynogenesis (equivalently sperm dependent parthenogenesis or sexual parasitism) is an asexual mode of reproduction in which paternal sperm is needed for activation of an egg and embryonic development, but sperm does not genetically contribute to the offspring, and the offspring is distributed clonally [73]. It is generally known that gynogenetic females produce unreduced eggs, i.e., eggs with the same ploidy as ploidy in somatic cells [74]. In *Carassius*, an embryonic development stimulated by sperm of a heterospecific sexual host gives rise to gynogenetic and clonal progeny consisting of females only; thus, unisexual populations arise [25]. An exception to the rules of unisexuality and asexuality has appeared by an occurrence of *Carassius* polyploid males in nature [75]. The exception has brought new enigmatic biological questions, such as the origin of polyploid *Carassius* males, or an incorporation or expression of paternal genetic material to a polyploid gynogenetic offspring with variable amounts of chromosomes [16]. Jiang et al. [75] proved a leakage of genetic information from heterospecific sperm into gynogenetic clones. An allogynogenetic offspring showed the effect of enhanced growth, skewed sex ratio, different liver isoenzyme patterns and variation in body color [75]. Allogynogenesis has become a cause of the occasional formation of a small proportion of males in gynogenetic offspring [76]. Significant incidence of natural triploid *Carassius* males has been up to 23% compared to the incidence of triploid females [52], but older literature introduced up to 10% [77]. Diploid *Carassius* males are common in nature with an occurrence of 1:1 for males and females within the diploid population [52]. Triploid males of *C. gibelio* have served as valuable experimental material for artificial crosses with the aim of finding out a mechanism of allogynogenesis/paternal leakage. If conspecific (i.e., from the same species) sperm from triploid *C. gibelio* inseminates an egg, the responding reproduction mode will be sexual with the generation of bisexual recombinant offspring. If heterospecific sperm from *Cyprinus carpio* activates *C. gibelio* egg development, the result is unisexual clonal lineage formed by gynogenesis [77,78]. A similar situation for all female progeny and gynogenesis would occur if sperm from goldfish initiates the egg and embryonic development of *C. langsdorfii* [79]. One small difference was evidenced in the development of *C. gibelio* [78] from the development of *C. langsdorfii* [79] eggs. An egg of *C. langsdorfii* retained somatic ploidy by retention of the first polar body [79] and an egg of *C. gibelio* extruded the first polar body, and the somatic ploidy level was restarted using extra DNA endoreplication without cytokinesis during the first mitosis in the stimulated egg [78]. Therefore, *C. auratus* complex demonstrates the coexistence of a dual mode of reproduction—sexual (recombinant) and asexual (gynogenetic) with the formation of unisexual (females only) and bisexual (both sexes) offspring [77].

## 6. Tell Me Your Ploidy and I Will Tell You Who You Are: Variability in Ploidy Levels and Chromosome Numbers

In general, it is possible to conclude that diploid *Carassius* females produce eggs with a reduced amount of genetic information in the nucleus. The spawning partners are conspecific males that fertilize eggs (true fertilization), and the resulted offspring is recombinant [16,80]. Triploid *Carassius* females can produce both gynogenetic and recombinant progenies dependent on heterospecific and conspecific sperm interaction, respectively [77]. Tetraploid *Carassius* females reproduce gynogenetically [44]. Exceptionally occurring tetraploid males have low motility spermatozoa, and thus poor fertilization capacity has been discovered [81]. Therefore, the knowledge of the ploidy level can tell us the origin of the examined *Carassius* individual (i.e., whether it is a clonal product of gynogenesis or result of recombination) and subsequent reproductive mode of the examined individual (in what mode its progeny will be produced). Therefore, determination of ploidy level in *Carassius* can serve as an additional marker helpful for species identification and important marker for determination of reproductive mode.

Artificial crossing experiments of *Carassius* individuals with different ploidy levels have been performed in order to find out variability or stability in chromosome numbers. Kalous and Knytl [16] performed a crossing experiment of diploid *C. gibelio* male with 100 chromosomes and triploid female with 159 chromosomes. Resulted F1 offspring possessed 150, 151, 156, 158, and 159 chromosomes. The mode of reproduction was not studied, but there was evident paternal leakage, and European *C. gibelio* showed high genome plasticity and diversity in chromosome numbers. Variation in the number of chromosomes of Asian *Carassius* was investigated by Zhou and Gui [39], who applied a crossbreeding experiment between triploid 156 chromosomal male and triploid 162 chromosomal female of Chinese *C. gibelio* and observed the effects in the F1 generation. Progeny contained and intermediate number of chromosomes (3n=6x=159) between paternal (3n=6x=156) and maternal (3n=6x=162) chromosomes. In addition, SCAR (sequence character amplified region) markers indicated recombination in offspring and originality of each parent from different hybridization events [39]. Both previous works indicate a close relationship between triploid karyotypes of European and East Asian *C. gibelio* populations, i.e., an ability to form a diverse number of chromosomes.

Variability in the number of chromosomes within *Carassius* has been previously described in multiple studies (e.g., [16,39,80], also see Table 1), including variability in diploid chromosome numbers (e.g., [82,83], reviewed in Knytl and Fornaini ([26], Table 3). In particular, these variations in diploidy are likely due to errors in karyotyping, which were commonly made during this period of early cytogenetic research [16,38]. Descriptions of triploids and tetraploids have ranged in chromosome number from 141 [25] to 166 [33] and from 200 [51] to 240 [84], respectively. Furthermore, variation in polyploid *Carassius* species are likely not biased by such errors in karyotyping and instead are thought to arise through mechanisms of allogynogenesis/paternal leakage [76,85]. Additional impacts on the variability in *Carassius* chromosome numbers can be attributed to an abundance of tiny chromosomes identified during karyotyping such as microchromosomes, also called B chromosomes. In particular, these B chromosomes are not easily counted and incorporated into the karyotype [16]. Despite these factors that make the characterization of chromosomes more difficult, we can conclude that the polyploid *Carassius* biotypes span a wide variation in chromosome number, with nuclei of diploid *Carassius* containing 100 chromosomes.

## 7. Fluorescent In Situ Hybridization vs. Cryptic Interspecies Hybridization

Various ploidy levels can be formed by two mechanisms: autopolyploidization and allopolyploidization. In autopolyploidization, new polyploid biotypes arise from a single parental ancestor. Environmental factors, such as heat shock, prevent the extrusion of the polar body in meiosis II and promotes polyploidization [86]. Environmental chemical factors, including the presence of bisphenol S, induces aberrant formation of spindle fibers, enabling the unequal division of chromatin during oogenesis [87]. Other chemical inducers of polyploidization include colchicine [88], which is widely used in the preparation of chromosomal suspensions, preventing chromosomal cleavage and the subsequent migration of each chromatid towards the centrosomes (e.g., [89,90]).

The second mechanism generating higher ploidy levels is allopolyploidization, when two or more distinct species hybridize. The precise mechanism of a heterospecific sperm genome incorporation into *Carassius* egg is unknown yet, but several factors, which influence chromosomal segregation or promote retention of the polar body during cell division, have been described [78].

In general, the asexual polyploid biotypes, which form unisexual populations, are considered to be of a hybrid origin [91], i.e., to be a product of allopolyploidization. *Carassius* parental species have not been identified, but the allopolyploid origin of *C. auratus* and *C. gibelio* has been recently revealed [92,93]. Interspecies hybridization in *Carassius* is a spontaneously ongoing process and several hybrids were identified in nature [21,94,95,96], and artificially allotetraploid hybrids were generated [71,72]. The exact ratio of maternal and paternal chromosomes within a hybrid genome was specified in naturally occurring *Carassius* allotetraploid by genomic in situ hybridization (GISH) [21]. Mitochondrial DNA confirmed *C. gibelio* maternal origin, and the GISH technique identified 50 *C. carassius* chromosomes out of 206 chromosomes of hybrid *Carassius* female (Figure 2).

Other allotetraploid *Carassius* hybrids were artificially generated and cytogenetically investigated. *Carassius* Hybrids with 206 and 212 chromosomes contained 156 and 162 *C. gibelio* chromosomes, respectively [71,72]. Both allotetraploids contained paternal 50 *Cyprinus carpio* chromosomes (2n=4x=100), which were incorporated into the maternal genome during fertilization. In all three cases [21,71,72], *Carassius* hybrids were produced by sperm genome addition to an egg of a gynogenetically reproducing triploid *Carassius* female.

Analysis of microsatellite loci, proposed by Hanlfling et al. [20], has been used for fast distinction of pure *Carassius* species from hybrids. Such techniques have successfully been applied for thorough genetic screening of natural *Carassius* populations, allowing for the identification of cryptic hybridization events between invasive and native taxa [95,96]. Interestingly, cryptic hybridization was proposed as one of the possible reasons for the declining occurrence of *C. carassius* [67,97]. Unfortunately, the ploidy level of naturally occurring cryptic *Carassius* hybrid has not been determined in previous studies [95,96], and we can only speculate if tetraploid *Carassius* biotypes were prevalent or exclusively present within these hybrids. Biochemical markers can also be used to identify the occurrence of cryptic hybridization, which is highly prevalent across diploid, triploid, and tetraploid ploidy levels. Tetraploid individuals have been exclusively represented by interspecific hybrids and no pure tetraploid *Carassius* biotype was identified [98]. This suggests that tetraploid *Carassius* biotypes are interspecific hybrids generated by sperm genome addition of sexual host species, usually the addition of a haploid 50-chromosome set into a triploid 150-chromosomal recipient. In addition, there is the question of whether asexual *Carassius* tetraploids are an evolutionary dead-end or whether rapid divergence is enabled through gynogenetical reproduction and the exploitation of “sexual parasitism”. Such allotetraploid hybrid species are capable of producing fertile allotetraploid gynogenetic offspring [72], which may enable rapid and effective distribution in nature.

Another way to distinguish maternal and paternal genome complements within hybrids is using FISH with whole chromosome painting probes. Individual chromosomes [99] or morphologically similar groups of chromosomes [58] can be separately dissected and used as a probe for in situ hybridization. The intensity of the fluorescent signal of painting probes can indicate evolutionary distance and distinguish parental subgenomes in biotypes that originate from interspecies hybridization [100]. B chromosomes can be easily distinguished and dissected from the *Carassius* karyotype. Yi et al. [76] dissected B chromosomes from metaphase spreads of allotriploid *C. gibelio* and detected eight positive signals on telomeric and pericentromeric regions of *Megalobrama amblycephala* (Yih, 1955) chromosomes. *Megalobrama amblycephala* was a parental species, and thus, Yi et al. [76] provided evidence of allogynogenetic introgression (paternal leakage) of the paternal *M. amblycephala* genome to the triploid *C. gibelio* maternal genome. No chromosomal microdissections have been carried out on naturally occurring *Carassius*, and it would be a prospective challenge for better understanding paternal leakage and determining the parental species of the polyploid *Carassius* biotypes.

## 8. Equilibrium between Quantitative Gynogenesis and Qualitative Sexuality (Red Queen Hypothesis)

The gynogenetic reproduction mode and production of all female populations without the need for coupling with conspecific males caused rapid invasivity and expansion of polyploid *Carassius* biotypes almost all over the world [12,94,101,102,103]. Excessive *Carassius* expansion entailed negative consequences due to successful competition, interspecies hybridization, and occupation of new areas [104,105]. The negative consequences are declining numbers or even extinction of the native ichthyofauna. After reaching a peak incidence in the population, the most common (clonal gynogenetic) phenotype is becoming more vulnerable to attacks by biological enemies than a far less frequent sexual population [106]. Herpesviral hematopoietic necrosis, caused by cyprinid herpesvirus-2, has affected many natural populations and commercial farms of gynogenetic *Carassius* biotypes [107,108,109,110,111]. The Red Queen hypothesis predicts evolution towards equilibrium in the populational composition of sexual and asexual biotypes coexisting together and co-evolving with parasites. In this coexistence, gynogenetic nonrecombinant clones form the most common phenotype. A sexual recombinant rare phenotype is favored and expected as more likely to escape infection by co-evolved parasites. This hypothesis was tested in sexually and gynogenetically coexisting fish populations of *Poecilia monacha* and *P. 2monacha-lucida*, respectively. Significantly higher accumulation of parasites was found in prevalent gynogenetic *P. 2monacha-lucida* [106]. Significantly higher parasite load was also found in asexual than in sexual *Carassius* biotypes. The less common sexual biotypes had higher immune reaction and resistance to parasites [112]. The coexistence of gynogenetic and sexual *Carassius* biotypes might be a trustworthy instance of the Red Queen hypothesis as a driver of *Carassius* biodiversity, but *C. gibelio* is still prevalent in natural waters, and thus, the equilibrium is not yet established by the Red Queen hypothesis. We propose the current relationship between co-occurring invasive and native *Carassius* as the birth of the existence of the Red Queen hypothesis.

## 9. Dual Mode of Sex Determination

Sex determination, i.e., whether an embryo develops into a male or female individual, can be governed environmentally by temperature (environmental sex determination) or genetically by a cascade of influential sex-determining genes located on sex chromosomes (genetic sex determination). Sex determination in fish is a diverse process with various mechanisms of both genetic and environmental sex determination (reviewed in [113]). *Carassius* is a typical example of both possible ways of sex determination. The genetic sex determination system is heterogametic for males (XY) and homogametic for females (XX). The XX/XY was evidenced in *C. auratus* using a crossbreeding experiment of hormone-treated (sex-reversed) with normal individuals followed by analysis of the progeny sex ratio [114]. Male heterogametic sex of *C. auratus* was identified cytogenetically using the C-banding method. Intensive heterochromatic blocks were found on the short arms of the second largest submetacentric chromosome pair in a diploid female and only one of these two heterochromatic blocks has been found in a diploid male. These cytogenetic findings confirmed the XX/XY sex determination system [8]. Zan [28] found a heteromorphic chromosome pair in a diploid *C. auratus* male, indicating male heterogamety and a far smaller Y chromosome than X. Additionally, Zan [28] identified three X chromosomes in a triploid *Carassius* female. However, C-banding revealed ten additional chromosomes, and there is no justified conclusion why three C-banded chromosomes were determined as three X chromosomes. Neither C-banding differences in number and position between sexes have been found in European *C. carassius* [49] nor in Japanese *Carassius auratus* complex [8]. Recently, Wen et al. [18] revealed a relatively large non-recombining region on the *C. auratus* Y chromosome. This sex region of the length of ∼11.7 Mb mapped on linkage group 22 involves male-specific genetic markers. Some of these markers were found in all XY males and were absent in all but one XX females/sex-reversed XX males. Even though sex-linked markers were tested on a less diverse *C. auratus* artificial population [18], the results will help to study the mechanisms of sex determination in a diverse *C. auratus* complex.

The other mechanism of sex determination in *Carassius* is given by the temperature of the environment in which the embryo develops [18,115,116]. Increasing of rearing temperature (usually 25 °C and higher) promotes a skewed sex ratio towards males. Sex reversal from female to male embryos has been detected after 12 days after fertilization [115], and sex-reversed phenotypic males were genotypic XX females/neomales [18]. Moreover, environmental temperature together with oxygen concentration are associated with the expression of *hypoxia-inducible transcription factor-1* gene, and this gene may be involved in sexual differentiation and an ability to display an extreme anoxia tolerance of *C. carassius* and *C. auratus* [117]. Apart from male and female sex, hermaphroditic individuals of *C. auratus* with both testes and ovaries were found in Western Siberia [118]. A natural occurrence of *Carassius* hermaphrodites has probably been caused by wide-ranged oscillation of water level, salinity and temperature. The Siberian area, where *C. auratus* hermaphrodites have been caught, is characterized by (1) long-lasting regression of water level, (2) high variability of water salinity and (3) fluctuation of water temperature in spring (spawning time of *Carassius*) from 10 °C at night to 30 °C in the day time [118]. No molecular cytogenetic and genetic investigations have been conducted to find out the hermaphroditic evolutionary history. The other question that remains open is whether factors (1), (2) and (3) synergically collaborate together or whether only one of these three factors is epistatically responsible for the natural production of hermaphroditic phenotype in *Carassius*. Temperature can likely influence hermaphroditism, and thus, the sex of *Carassius* can be skewed towards one favorable sex.

## 10. Perspective of a Future Research

Hybridization has often been viewed as a destructive force leading to hybrid sterility or non-viability, but it is increasingly being recognized as a potential creative force in evolution. Novel hybrids can benefit from rapid adaptation to new environmental conditions and the potential to exploit ecological niches where a primary reproductive barrier would have existed between hybrids and parental species [119,120]. For example, *Carassius* benefit from the establishment of new genome combinations with the potential for clonal reproduction [120,121]. In accordance with the hybrid speciation hypothesis [122], novel hybrid species can be created after successive rounds of hybridization occur [67]. Cytogenetic analysis can identify parental genomes and reveal their exact ratio of subgenomes within the hybrid genome, yet parental lineages remain difficult to distinguish after several rounds of hybridization despite extensive investigation of *Carassius* ancestors.

Differing modes of reproduction are natural phenomena, which are hypothesized to be associated with the transition between genetic and environmental sex determination [123]. The use of a cytogenetic, genomic, and gene editing framework can be of particular use to study the transition between mechanisms of sex determination and modes of reproduction. Next-generation sequencing (e.g., restriction site-associated DNA sequencing, RNA sequencing) is widely used in the identification of non-recombining regions, sex-specific single nucleotide polymorphism (SNPs) and/or differentially expressed genes [18,124,125,126]. Thus far, the field has characterized the non-recombining genomic regions in artificial population, but the identification of master sex-determining genes or regulators on the top of the developmental cascade has not yet been elucidated.

Once identified, FISH can be used to map the physical location of sex-determining genes on chromosomes [127,128]. Furthermore, the functionality and respective non-functionality of a master sex-determining gene can be described in vitro using modern molecular tools, such as Clustered Regularly Interspaced Short Palindromic Repeats (CRISPR) with Cas9 protein (cleaves the targeted genomic region and introduces small indels that cause premature stop codons), enabling gene knockout experiments. Due to the activation of a compensatory mechanism against deleterious mutations, CRISPR/Cas9 knockout of some genes may not produce a measurable change in phenotype. In such cases, alternatives to CRISPR may be useful, including morpholino, which binds to a target RNA and inhibits protein synthesis [129]. One case of successful CRISPR/Cas9 knockout in *C. cuvieri* and *C. cuvieri* x *C. auratus* hybrids was where melanin synthesis was suppressed via the disruption of a *tyrosinase* gene [130]. CRISPR/Cas9 knockout experiments were shown as a powerful tool for genetic engineering in aquaculture [130], and such genetic engineering techniques could be used to successfully knockout candidate sex-determining *Carassius* genes. Thus far, a single candidate master sex-determining gene has been proposed, the *anti-müllerian hormone* gene (*amh*) [18]. Furthermore, CRISPR/Cas9 could be a prospective tool for testing the functionality of genetic mechanisms hidden beyond clonality and/or polyploidy as has been similarly suggested by Yin et al. [131].

## 11. Conclusions

*Carassius* is a highly diverse complex involving several taxa with different mitochondrial lineages [12], several ploidy levels [25], dual mode of reproduction, unisexual and bisexual populations [17,77], both genetic and environmental modes of sex determination [18], interspecific hybrids [21] and hermaphrodites [118]. In this review, we highlighted *Carassius* as an experimental fish model to study the evolution of the above-listed unusual and incredible phenomena using already well-established molecular cytogenetics coupled with perspective genomics and genetic engineering.

In Europe the genus *Carassius* involves two diametrically distinct entities: the first one is invasive and gynogenetic *C. gibelio*, and the second one is vulnerable sexual *C. carassius*. In the wild, *C. carassius* is rapidly becoming an endangered species in many regions [20,67,97,104], due to competition from other *Carassius* species, such as *C. gibelio*, which has colonized novel ecological niches enabled by the mode of asexual reproduction, avoiding the costs associated with sexual reproduction. In order for conservation efforts to be directed to pure *C. carassius* lineages [104,132], it is necessary to identify individuals with molecular cytogenetic techniques (i.e., karyotype analysis, in situ hybridization). Determination of ploidy levels can aid in taxonomic identification of *Carassius* (Section 6), and allow for reliable discrimination between pure species from gynogenetic biotypes/cryptic hybrids. We propose that cytogenetic investigations would not only help to protect the endangered *C. carassius* but also to understand the evolution of the highly diverse *Carassius* complex.

## Figures and Tables

**Figure 1 ijms-23-08095-f001:**
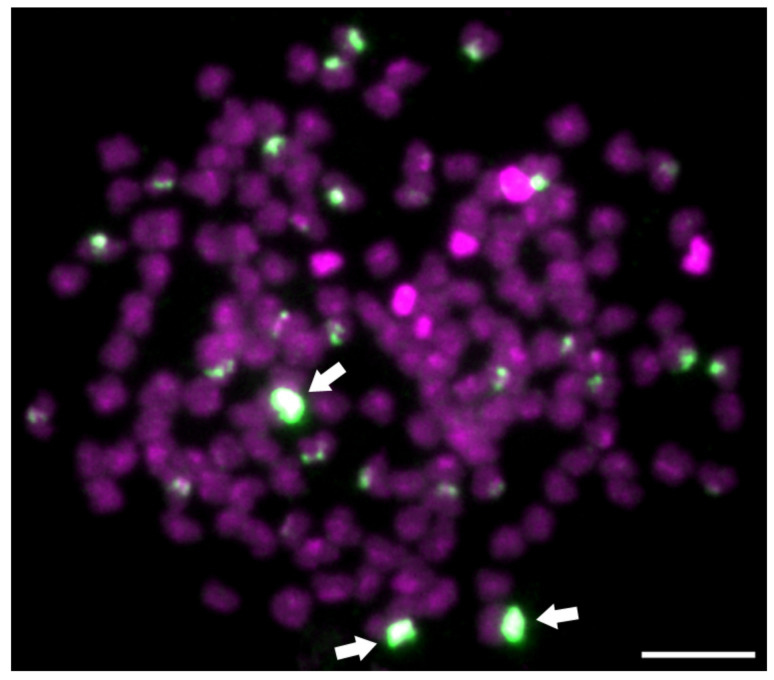
Fluorescent in situ hybridization (FISH) with *5S* ribosomal probes on *Carassius* hybrid female with 156 chromosomes. Three highly intensive *5S* gene loci in green (arrowheads) out of twenty-seven show triploid origin of this female. Chromosomes are counterstained with 4’,6-diamidino-2-phenylindole (DAPI) in blue and red. Scale bar represents 10 μm. Figure was modified according to Knytl et al. [67].

**Figure 2 ijms-23-08095-f002:**
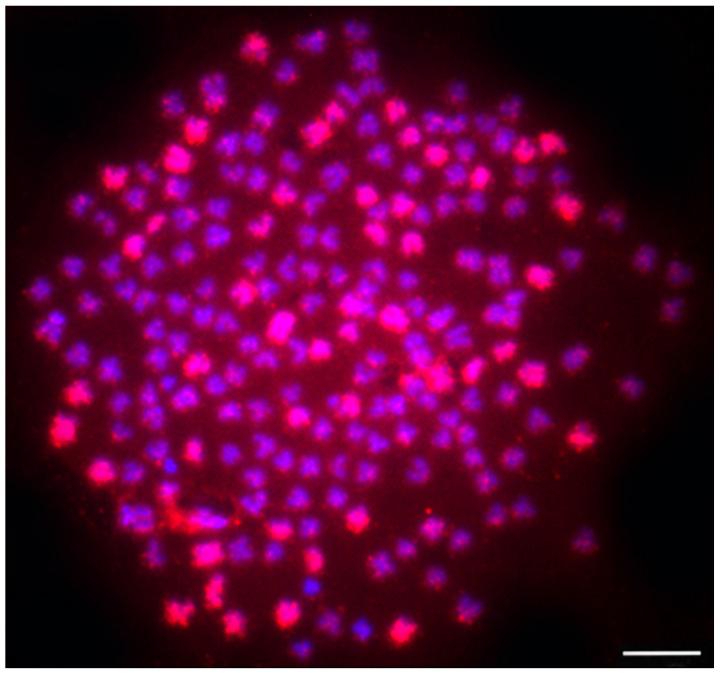
Genomic in situ hybridization (GISH) identified a wild *Carassius* hybrid female with 206 chromosomes. The 50 paternal (*C. carassius*) chromosomes are labeled with the whole genome painting probe, shown here in red. The other 156 maternal chromosomes (*C. gibelio*) are labeled by DAPI, in blue. Scale bar represents 10 μm. Figure was modified according to Knytl et al. [21].

**Table 1 ijms-23-08095-t001:** Previously published karyotype formulas within *C. auratus* complex, including information about sex and locality of the investigated individuals. Karyotype data are ordered chronologically. *Carassius carassius* and diploid *C. auratus* and *C. gibelio* are not involved. 2n = diploid, 3*n* = triploid, 4*n* = tetraploid, *m* = metacentric, sm = submetacentric, st = subtelocentric, *a* = acrocentric, *B* = B chromosome/microchromosome, NA = information not available, F = female, M = male.

Species, Ploidy, and Karyotype	Sex	Locality	Reference
*Carassius auratus* (goldfish)			
3n=6x=162 (33m+53sm+76st-*a*)	F	China	[27,28]
3n=6x=156 (30m+46sm+80st-*a*)	NA	China	[29]
3n=6x=162 (36m+56sm+70st-*a*)	NA	China	[30]
*Carassius gibelio*			
3n=6x=141	F	Belarus	[25]
3n=6x=156 (34m+62sm+60a)	NA	River Amur	[31]
3n=6x=150	F	Bosnia	[32]
3n=6x=166 (46m+82sm-st+32a+6B)	F	Czechoslovakia	[33]
3n=6x=160 (15m+28sm+126st-a+1B)	F	Yugoslavia	[34]
3n=6x=160	F	Czechia	[35]
3n=6x=162 (32m+52sm+78st-*a*)	NA	NA	[29]
3n=6x=158 (36m+54sm-st+68a)	F	Yugoslavia	[36]
3n=6x=156 (42m+74sm+40st)	F, M	China	[37]
3n=6x=150 (26m+50sm+74st-*a*)	F	Poland	[38]
3n=6x=156 (36m+54sm+60st-a+6B)	M	China	[39]
3n=6x=162 (42m+54sm+60st-a+6B)	F		
3n=6x=154 (24m+54sm+72st-a+4B)	F		
3n=6x=160 (33m+48sm+75st-a+4B)	F	Poland	[40]
3n=6x=160 (34m+58sm+62st-a+6B)	M		
3n=6x=156 (30m+54sm+66st-a+6B)	F	Czechia	[21]
*Carassius langsdorfii*			
2n=4x=100 (20m+40sm+40a)	F, M		
3n=6x=156 (34m+62sm+60a)	F	Japan	[41]
4n=8x=206 (44m+82sm+80a)	F		
3n=6x=156 (34m+62sm+60a)	F	Japan	[42]
2n=4x=100 (20m+40sm+40a)	F, M		
3n=6x=157 (32m+59sm+62a+4B)	F	Japan	[43]
3n=6x=165 (44m+82sm+80a+9B)	M		
3n=6x=156 (34m+62sm+60a)	F	Japan	[44]
4n=8x=206 (44m+82sm+80a)	F		
3n=6x=156	F	Czechia	[45]
*Carassius cuvieri*			
2n=4x=100 (12m+36sm+52a)	F, M	Japan	[46]
2n=4x=100 (20m+40sm+40a)	F, M	Japan	[43]
2n=4x=100 (12m+36sm+52a)	F, M	Japan	[47]
2n=4x=100 (12m+36sm+52st-*a*)	F, M	Japan	[48]
*Carassius buergeri*			
2n=4x=100 (20m+40sm+40a)	F, M	Japan	[42]
3n=6x=156 (34m+62sm+60a)	F		
2n=4x=100 (12m+36sm+52st-*a*)	F, M	Japan	[48]
*Carassius grandoculis*			
2n=4x=100 (20m+40sm+40a)	F, M	Japan	[42]
2n=4x=100 (12m+36sm+52st-*a*)	F, M	Japan	[48]

## Data Availability

Not applicable.

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
