# Peer review of "A Fish of Multiple Faces, Which Show Us Enigmatic and Incredible Phenomena in Nature: Biology and Cytogenetics of the Genus *Carassius"

_ijms, 2022, doi:10.3390/ijms23158095_

Round 1

Reviewer 1 Report

Knytl et al reviews characteristics in cytogenetics of the Genus Carassius. The information presented in this review is interesting, but I feel that several issues can be addressed to improve the manuscript.

1. I guess that numbers in glossary are references. If yes, should they be cited using the same style as in the main text?

2. In the legend to figure 1, for the scale bar, it should be 10 µm, not 10 µM, which represents the concentration.

3. For sections 5 and 6, it would be more straightforward and helpful to the readers if the authors could provide a table summarizing the information on the mode of reproduction, ploidy, chromosome numbers, and development of the genus Carassius. 

4. Line 335, “Carassius is an unique example of both possible ways of sex determination”. It seems for me that sex determination in many other fish can be also influenced environmentally and genetically. 

5. Lines 404-406, “Gene knockout using Clustered Regularly Interspaced Short Palindromic Repeats (CRISPR) with Cas9 protein editing approach can reveal functionality respective non-functionality of master sex determining gene”. This statement is somewhat misleading. There is plenty of evidence that knockout of many key genes does not produce any phenotype mostly due to activation of a genetic compensatory mechanism (for example, see Rossi et al., 2015). Therefore, it cannot be concluded that the absence of phenotypes reveals non-functionality of a gene. 

6. Many sentences need rephrasing, for example, “However, much contention surrounding the taxonomic classification of this species”,  “Variability in number of chromosomes in Asian Carassius confirmed Zhou and Gui”, etc. In many places, the wording is also inaccurate or inappropriate.

Reviewer 2 Report

The article titled: “A Fish of Multiple Faces, which Show Us Enigmatic and Incredible Phenomena in Nature: Biology and Cytogenetics of the Genus Carassius” is a review regarding the biology and cytogenetic characterizes of the Carassius genus as useful taxum in order to study many evolutionary biological and genetics factors. Particularly the focus is also on the cytogenetic approach useful for species identification. The work is interesting and worthy of publication but minor points need to be more defined; furthermore, a special attention need to be put forward to the cytogenetic approach because this part need to be more exhaustive especially in the literatures section.

Line 3, 4

you speak about two entities. But however you are making a revision on all the group, thus I would avoid this sentences. Or you can say that among the others a special attention would be put forward these two entities.

Line 68-81

you write a new genus name Cyprinus gibelio but without introducing it. Please better explain.

Line 116- 130

better rewrite this section. Since a special attention has to be focus on the cytogenetic approach it is necessary to better introduce it.

Line 117 Regarding “banding techniques have served as useful tool for efficient and precise analyses of karyotypes in various animals” (add literature as for example: Dumas, F.; Sineo, L.; Ishida, T. Taxonomic identifcation of Aotus (Platyrrhinae) through cytogenetics. J. Biol. Res 2015, 88, 65- 542 66. 543 52.

Line 119-123 place together with line 129-130- both parts regard the molecular cytogenetic approach

Line 129-130 “various types of targeted nucleotide sequences”- add used as probes in various animals and in Line 130 Add more references as suggested
